# A CNOT gate between multiphoton qubits encoded in two cavities

S. Rosenblum[1,2], Y.Y. Gao[1,2], P. Reinhold [1,2], C. Wang [1,2,4], C.J. Axline[1,2], L. Frunzio [1,2], S.M. Girvin [1,2], Liang Jiang [1,2], M. Mirrahimi[2,3], M.H. Devoret[1,2] & R.J. Schoelkopf[1,2]

Entangling gates between qubits are a crucial component for performing algorithms in quantum computers. However, any quantum algorithm must ultimately operate on error-protected logical qubits encoded in high-dimensional systems. Typically, logical qubits are encoded in multiple two-level systems, but entangling gates operating on such qubits are highly complex and have not yet been demonstrated. Here we realize a controlled NOT (CNOT) gate between two multiphoton qubits in two microwave cavities. In this approach, we encode a qubit in the high-dimensional space of a single cavity mode, rather than in multiple two-level systems. We couple two such encoded qubits together through a transmon, which is driven by an RF pump to apply the gate within 190 ns. This is two orders of magnitude shorter than the decoherence time of the transmon, enabling a high-fidelity gate operation. These results are an important step towards universal algorithms on error-corrected logical qubits.

[1] Department of Applied Physics and Physics, Yale University, New Haven, CT 06520, USA. [2] Yale Quantum Institute, Yale University, New Haven, CT 06520, USA. [3] QUANTIC team, INRIA de Paris, 2 Rue Simone Iff, 75012 Paris, France. [4] Present address: Department of Physics, University of Massachusetts, Amherst, MA 01003, USA. S. Rosenblum and Y. Y. Gao contributed equally to this work. Correspondence and requests for materials should be addressed to S.R. (email: serge.rosenblum@yale.edu) or to R.J.S. (email: robert.schoelkopf@yale.edu)

I n traditional approaches to quantum error correction, bits of quantum information are redundantly encoded in a register of two-level systems[1, 2]. Over the past years, elements of quantum error correction have been implemented in a variety of platforms, ranging from nuclear spins[3], photons[4], and atoms[5], to crystal defects[6] and superconducting devices[7–9]. However, for performing actual algorithms with an error-protected device, it is necessary not only to create and manipulate separate logical qubits, but also to perform entangling quantum gates between them. To date, a gate between logical qubits has not yet been demonstrated, in part due to the large number of operations required for implementing such a gate. For example, in the Steane code[2,10], which protects against bit and phase flip errors, a standard logical CNOT gate would consist of seven pairwise CNOT gates between two seven-qubit registers[11]. Previous experiments have demonstrated an effective gate between two-qubit registers that are protected against correlated dephasing[12]. In that case, an entangling gate could be implemented using just a single pairwise CNOT gate between the registers.

We choose to pursue a different strategy by encoding qubits in the higher-dimensional Hilbert space of a single harmonic oscillator[13], or more concretely in multiphoton states of a microwave cavity mode[14,15]. This approach has the advantage of having photon loss as the single dominant error channel, with photon-number parity as the associated error syndrome. Codes whose basis states have definite parity, such as the Schrödinger cat code[16] or the binomial kitten code[17], can then be used to actively protect quantum information against this error[8,18]. While preparation of an entangled state between two cavities has been performed before[19,20], a quantum gate between two multiphoton qubits has so far not been demonstrated. In contrast to gates between two-level systems, which can be coupled by a linear element such as a cavity bus[21], harmonic oscillators can nontrivially interact only if they are coupled by a nonlinear ancillary element. However, the requirement for fast interaction between the cavities without inheriting large undesired nonlinearities and decoherence from the ancilla, presents a challenge to the cavity-based approach to quantum error correction.

In this work, we address this challenge by coupling the two cavities to an RF-driven ancilla transmon. The cavities interact sequentially with the ancilla to effectively implement a CNOT gate between the two encoded multiphoton qubits. We generate a high-fidelity multiphoton Bell state, perform quantum process tomography of the gate, and apply the gate repeatedly in order to quantify imperfections in the operation. The number of CNOT operations that we can coherently apply is $\sim 10^2$, bringing this gate within the regime required for practical quantum operations[22,23]. We also measure the undesired entangling rate between the cavities during idle times, and infer a high on/off ratio of the entangling rate[24,25] of $\sim 300$. This figure of merit is important since undesired cross-talk is often a major hurdle when trying to scale up to a larger number of qubits.

## Results

### System description.
The experimental system used for implementing the entangling gate is depicted in Fig. 1. The multiphoton qubits are encoded in two high-Q ($T_1 \sim 0.002$ s) superconducting coaxial stub cavities. Several different multiphoton encodings are compatible with the entangling gate (see Supplementary Note 7). Here we choose a basis of even-parity Fock states

$$|0_L\rangle_C = |0\rangle_C, \quad |1_L\rangle_C = |2\rangle_C \tag{1}$$

for the control cavity, and Schrödinger kitten states[17]

$$|0_L/1_L\rangle_T = \frac{1}{\sqrt{2}}\left(\frac{|0\rangle_T + |4\rangle_T}{\sqrt{2}} \pm |2\rangle_T\right) \tag{2}$$

for the target cavity (henceforth omitting normalization). These encodings can allow error detection of a photon loss event in both cavities, as well as error correction in the target cavity.

The operation of the gate relies on two types of nonlinear interaction between the cavities and the ancilla, enabled by the ancilla's Josephson junction. The first is the naturally occurring dispersive interaction, which can be understood as a rotation of the cavity phase space conditioned on the ancilla state. Here we consider the ancilla ground and second excited states $|g\rangle$ and $|f\rangle$ only, since the first excited state $|e\rangle$ is ideally not populated during the gate operation. In this case the Hamiltonian is

$$\hat{H}_{\rm disp}/\hbar = -\tilde{\chi}_T \hat{a}_T^\dagger \hat{a}_T |f\rangle\langle f| - \tilde{\chi}_C \hat{a}_C^\dagger \hat{a}_C |f\rangle\langle f|, \tag{3}$$

where $\hat{a}_{C(T)}$ is the control (target) annihilation operator. As a result of this interaction, the target (control) cavity phase space rotates at $\tilde{\chi}_{C(T)}/2\pi = 1.9$ MHz (3.3 MHz) when the ancilla is in $|f\rangle$, but remains unchanged when the ancilla is in $|g\rangle$.

We can also drive a sideband interaction between the control cavity and the ancilla using an RF tone that satisfies the frequency matching condition $\omega_p = \omega_{gf} - \omega_C - (n_C - 1)\tilde{\chi}_C$, with $\omega_{gf}/2\pi = 9.46$ GHz the ancilla transition frequency between $|g\rangle$ and $|f\rangle$ (Fig. 2b), and $n_C$ the number of control photons (we discuss the effect of the target photon number $n_T$ in Supplementary Note 4). This interaction, described by the Hamiltonian

$$\hat{H}_{\rm sb}/\hbar = \frac{\Omega_C(t)}{2}\left(\hat{a}_C |f\rangle\langle g| + \hat{a}_C^\dagger |g\rangle\langle f|\right), \tag{4}$$

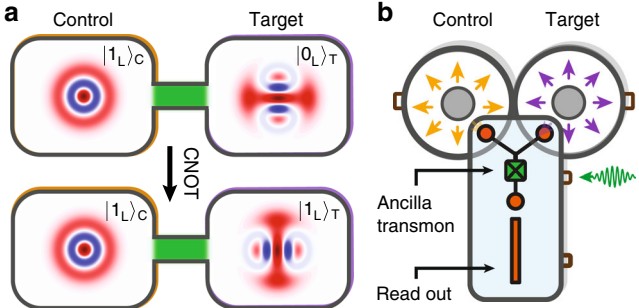

**Fig. 1** Experimental implementation of an entangling gate between multiphoton qubits encoded in two cavities. **a** Example of the CNOT operation. In the initial state, illustrated by the Wigner distributions in the top panel, the control qubit is in $|1_L\rangle_C$, and the target qubit in $|0_L\rangle_T$ (as defined in Eqs. (1) and (2)). Under the action of the CNOT gate, enabled by a nonlinear coupling between the cavities (in green), the target state at the output (bottom panel) is inverted to $|1_L\rangle_T$. **b** Sketch of the device, which is housed inside an aluminum box, and cooled down to 20 mK. The control and target qubits are encoded in photon states of the fundamental modes (yellow and purple arrows) of two coaxial cavities with frequencies $\omega_C/2\pi = 4.22$ GHz and $\omega_T/2\pi = 5.45$ GHz, respectively. The ancilla transmon ($\omega_q/2\pi = 4.79$ GHz) has two coupling pads (orange circles) that overlap with the cavity fields. Cavity-ancilla interaction is achieved by application of a frequency-matched RF drive (green arrow) to the coupling pin near the Josephson junction (marked by X). The ancilla also serves to prepare and read out the cavity state, and is measured by its dispersive coupling to a stripline readout resonator (orange rectangle). More details on this device can be found in Supplementary Note 1 and ref. [20]

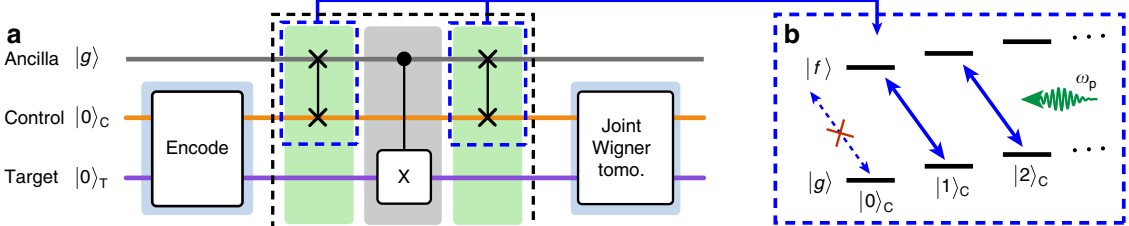

**Fig. 2** Protocol of the entangling gate. **a** The sequence starts with preparation of the desired initial two-cavity state, while leaving the ancilla transmon in the ground state. The cavity–cavity CNOT gate (dashed black rectangle) consists of two entangling gates between the control cavity and the ancilla (dashed blue rectangles), interleaved by a CNOT gate between the ancilla and the target, implemented by a conditional $\pi/2$ phase-space rotation of the target cavity. The joint Wigner distribution of the final two-cavity state is measured using a method similar to ref. [20]. **b** Schematic level diagram illustrating the RF-driven control-ancilla sideband transition. Through the absorption of a single drive photon (in green) and a single control photon, the ancilla is doubly excited from $|g\rangle$ to $|f\rangle$ (solid blue arrows). However, when the control cavity is in vacuum, the absence of a control photon prevents the ancilla from being excited to $|f\rangle$ (dashed blue arrow)

leads to sideband oscillations[26] between the states $|n_C, g\rangle$ and $|n_C - 1, f\rangle$[27–29]. By strongly driving this transition we obtain an oscillation rate of $\sqrt{n_C}\Omega_C/2\pi = 11.2$ MHz with $n_C = 2$, close to the theoretical prediction (see Supplementary Note 3). However, for $n_C = 0$ the pump does not drive sideband oscillations, and the ancilla remains in its ground state (Fig. 2b).

**Gate protocol.** The basic mechanism behind the gate is to make the cavities interact sequentially with the ancilla, enabling an effective nonlinear interaction between the cavities without requiring a significant direct cavity–cavity coupling. This method is similar to the one used in a recent experiment on a gate between single optical photons[30]. We start by preparing the desired initial state using optimal control pulses on the ancilla and on the cavities[31–33], after ensuring that the ancilla is initialized in $|g\rangle$. The gate sequence itself is then performed in three steps (Fig. 2a). First, we apply the sideband drive for $\pi/(\sqrt{2}\Omega_C) = 45$ ns, exciting the ancilla to $|f\rangle$ conditioned on the control being in $|1_L\rangle_C$. We then turn off the drive for 100 ns ($\sim\pi/2\tilde{\chi}_T$, see Supplementary Note 4), during which the ancilla dispersively interacts with the target cavity. This flips $|0_L\rangle_T$ into $|1_L\rangle_T$ and vice versa, conditioned on the ancilla being in $|f\rangle$. We then apply the sideband drive a second time to disentangle the ancilla from the cavities, thereby effectively achieving a CNOT gate between the two cavities after a total gate time of $t_g \sim 190$ ns. Finally, we use the ancilla to perform joint Wigner tomography on the two-cavity state[20,34], from which we can reconstruct the density matrix (see Supplementary Note 6).

**Gate characterization.** The hallmark of a CNOT gate is its ability to entangle two initially separable systems. As a demonstration of this capability, we apply the gate to $|\psi_{in}\rangle = (|0_L\rangle_C + |1_L\rangle_C) \otimes |0_L\rangle_T$ (Fig. 3a). Ideally, this should result in a logical Bell state $|\psi_{ideal}\rangle = |0_L\rangle_C|0_L\rangle_T + |1_L\rangle_C|1_L\rangle_T$. By reconstructing the output density matrix $\rho_{meas}$ (Fig. 3b), we deduce a state fidelity of $F_{Bell} \equiv \langle\psi_{ideal}|\rho_{meas}|\psi_{ideal}\rangle = (90 \pm 2)\%$. This is within the measurement uncertainty of the input state fidelity $F_{in} = (92 \pm 2)\%$. Therefore, we conclude that the effect of nonidealities in the gate operation on the Bell state fidelity is obscured by imperfections in state preparation and measurement (see Supplementary Note 5).

To fully characterize the CNOT gate, we next perform quantum process tomography[35] (QPT). We achieve this by applying the gate to sixteen logical input states that together span the entire code space. By performing quantum state tomography on the resulting output states we can reconstruct the quantum process $\epsilon(\rho_{in})$, which captures the action of the gate on an

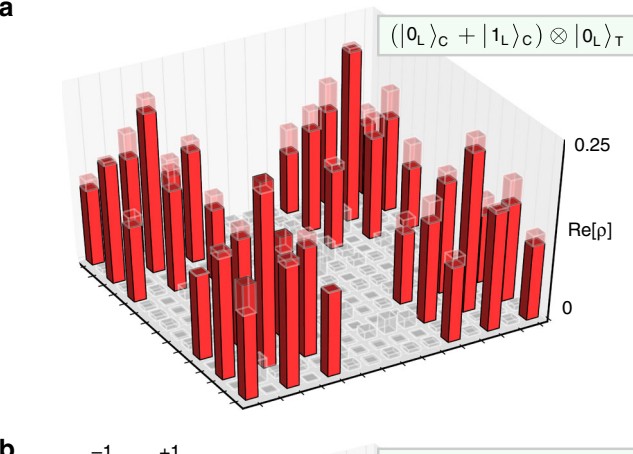

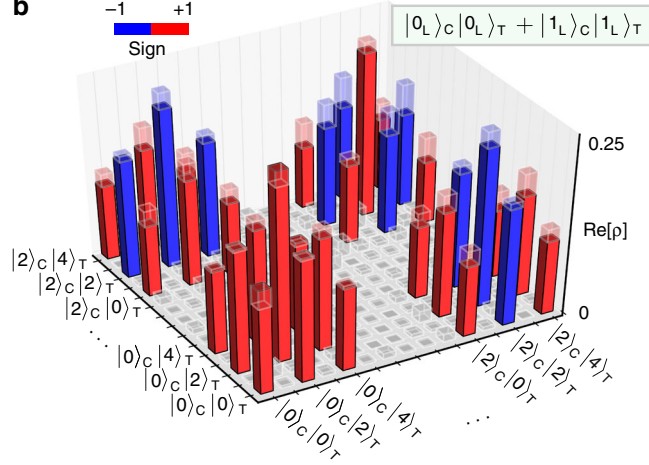

**Fig. 3** Generation of a multiphoton Bell state. Reconstructed density matrices (solid bars) of **a**, the initial separable two-cavity state $(|0\rangle_C + |2\rangle_C) \otimes \left(\frac{|0\rangle_T + |4\rangle_T}{\sqrt{2}} + |2\rangle_T\right)$ (ideal shown by transparent bars) and **b**, the output state after application of the CNOT gate, turning the kitten state into $\left(\frac{|0\rangle_T + |4\rangle_T}{\sqrt{2}} - |2\rangle_T\right)$, provided the control state is $|2\rangle_C$. We reconstruct the density matrices assuming a Hilbert space spanned by the Fock states $|n\rangle_C|m\rangle_T$ with $n < 3$ and $m < 5$ after confirming the absence of population at higher levels. Components of the density matrices below 0.05 are colored in gray for clarity. The imaginary parts are small as well, and are shown in Supplementary Note 8 for completeness

arbitrary input state $\rho_{in}$. The result can be expanded in a basis of two-qubit generalized Pauli operators $E_i$ on the code space as $\epsilon(\rho_{in}) = \sum_{m,n=0}^{15} \chi_{m,n} E_m \rho_{in} E_n$, where $\chi$ is the process matrix. Using the measured $\chi$ (Fig. 4a), we determine a process fidelity of $F_{CNOT} \equiv \mathrm{Tr}\{\chi_{ideal}\chi\} = (89 \pm 2)\%$. We can estimate the effect of nonideal state preparation and measurement by performing QPT on the process consisting of encoding and measurement only, yielding a fidelity with the identity operator of $F_{identity} = (92 \pm 2)\%$.

To more accurately determine the performance of the gate and highlight specific error mechanisms, we apply it repeatedly to various input states (Fig. 4b). We then measure how the state fidelity decreases with the number of gate applications. A first observation is that no appreciable degradation in state fidelity occurs when the control qubit is in $|0_L\rangle_C$. Indeed, the control cavity contains no photons in this case, and as a result the ancilla

remains in its ground state at all times. When the initial two-cavity state is $|1_L\rangle_C|X_L^-\rangle_T$ (introducing $|X_L^{\pm}\rangle \equiv (|0_L\rangle \pm |1_L\rangle)$ and $|Y_L^{\pm}\rangle \equiv (|0_L\rangle \pm i|1_L\rangle)$), corresponding to $|2\rangle_C|2\rangle_T$ in the Fock-state basis, the ancilla does get excited to the $|f\rangle$-state, and we measure a small decay in fidelity of $(0.6 \pm 0.3)\%$ per gate application. This is consistent with the ancilla decay time from $|f\rangle$ to $|e\rangle$, measured to be 40 μs. While the qubit is irreversibly lost when a decay occurs, the final ancilla state is outside the code space, and therefore this is a detectable error. If the control qubit is initially in a superposition state, the first sideband pump pulse will entangle the control cavity with the ancilla, making the state prone to both ancilla decay and dephasing ($T_2^f = 17$ μs). For example, for $|X_L^+\rangle_C|X_L^-\rangle_T$, we measure a decay in fidelity of $(0.9 \pm 0.2)\%$ per gate. When the target cavity state is not rotationally invariant (i.e. not a Fock state), we observe larger decay rates ($(2.0 \pm 0.3)\%$ for $|1_L\rangle_C|0_L/1_L\rangle_T$, and $(2.1 \pm 0.2)\%$ for $|Y_L^+\rangle_C|Y_L^+\rangle_T$). Possible mechanisms for these increased decay rates are discussed in Supplementary Note 4. While an accurate determination of the gate fidelity would require randomized benchmarking[36], the data presented in Fig. 4b is sufficient to infer an average degradation in state fidelity of ~1% per gate application, close to the ~ 0.5% limit set by ancilla decoherence.

An important figure of merit for an entangling gate is the ability to turn off the interaction, to prevent unwanted entanglement between the cavities. In practice, the cross-Kerr interaction between the cavities, described by the Hamiltonian $\hat{H}_{CT}/\hbar = \chi_{CT} \hat{a}_C^\dagger \hat{a}_C \hat{a}_T^\dagger \hat{a}_T$, induces entanglement even when the gate operation is not applied. To measure the interaction rate $\chi_{CT}$, we prepare a separable two-cavity state in a code space spanned by vacuum and the single-photon Fock state (Supplementary Note 7), and perform state tomography after variable delay times. When extracting the concurrence[37] of the measured density matrices (Fig. 5), we observe first an increase, then a subsequent decrease, of the entanglement between the cavities.

In a similar vein, when starting with a Bell state, the cross-Kerr interaction first disentangles, and then re-entangles the two cavities. The cavity dephasing times of ~ 500 μs lead to a gradual overall loss of entanglement in both cases. From the measured curves, we infer a cross-Kerr interaction rate of $\chi_{CT}/2\pi = 2$ kHz.

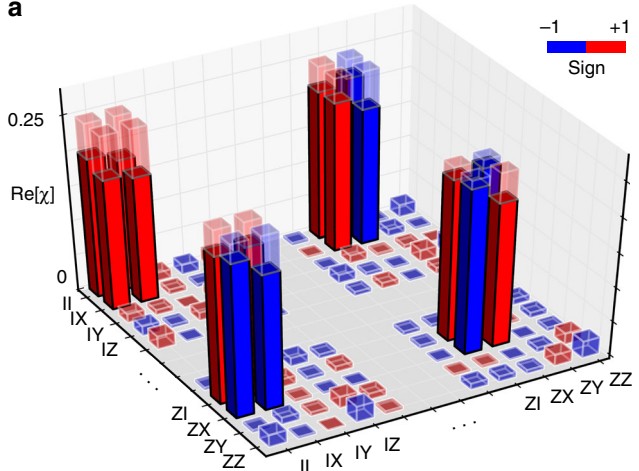

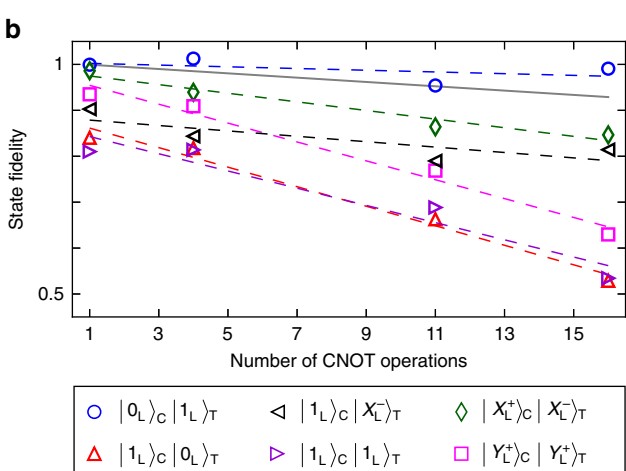

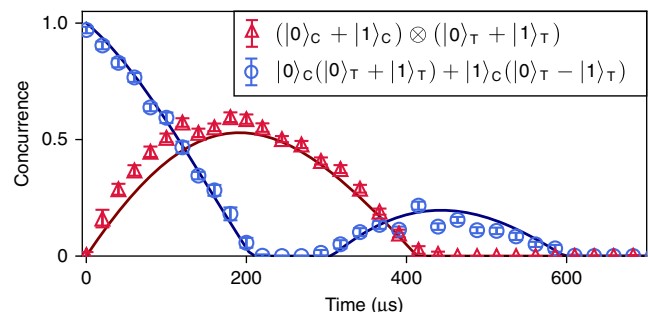

**Fig. 4** Characterization of the controlled NOT gate. **a** Quantum process tomography. The solid (transparent) bars represent the measured (ideal) elements of the process matrix $\chi$. The corresponding process fidelity is $F_{CNOT} = (89 \pm 2)\%$. For clarity, only the corners of the process matrix are presented. The full $\chi$-matrix is shown in Supplementary Note 8 for completeness. **b** State fidelity under repeated gate applications for various input states, chosen to highlight different error mechanisms of the gate (the dashed lines are linear fits). The solid gray line depicts the simulated average slope of state fidelity imposed by ancilla decoherence. The state fidelities are calibrated by the value measured for the vacuum state (Supplementary Note 6). The standard errors are derived from bootstrapping, and are equal in size to the symbols

**Fig. 5** Undesired entanglement induced by the coupling ancilla. Concurrence vs. wait time for an initially separable state (red) using single-photon encoding, and for an initial Bell state (blue) obtained by applying the CNOT gate to the separable state. The presence of the cross-Kerr interaction between the two cavities is responsible for the observed oscillatory behavior, whereas dephasing due to thermal excitations in the ancilla results in a gradual decay of the entanglement. By fitting simulations (solid curves) to the measured data, we determine a cross-Kerr interaction rate of $\chi_{CT}/2\pi = 2$ kHz. Error bars indicate the standard error derived from bootstrapping

However, the residual entanglement rate for the multiphoton encoding is increased to $\Omega_{\text{res}} = n_C \bar{n}_T \chi_{\text{CT}} = 2\pi \times 8\,\text{kHz}$, where $\bar{n}_T = 2$ is the average number of photons in the target cavity. We can therefore infer the on/off ratio of the entangling rate, defined by the ratio of the times to generate maximal entanglement without and with gate application, to be $\pi/(\Omega_{\text{res}} t_g) \sim 300$.

## Discussion

In conclusion, we have realized a high-fidelity entangling gate between multiphoton states encoded in two cavities. Together with single-qubit gates[33], this provides a universal gate set on encoded qubits that can be actively protected[8] against single-photon loss. The gate relies on correct operation of the control-ancilla sideband drive, restricting the choice of control encodings. In fact, the encoding used in this demonstration, as well as a variety of similar encodings compatible with the CNOT gate, provides full error-correctability for the target cavity, but only detectability of a photon loss error in the control cavity. However, a generalization of the kitten code exists which could potentially allow for identical error-correctable encodings in both cavities (Supplementary Note 7). An important criterion of a gate on error-corrected logical qubits is whether errors before or during the gate operation can be detected or corrected. Using our scheme, ancilla or cavity decay events can be detected since they lead to a final state outside the code space. However, the gate fidelity is ultimately limited by ancilla dephasing, and with the current encoding the control cavity is subject to uncorrectable no-jump evolution[17]. These remaining imperfections need to be addressed in future fault-tolerant gate implementations. The demonstrated gate is especially useful for practical applications that are limited by decoherence processes or spurious interactions during long idle times. In particular, it establishes the potential of multicavity registers for distributed quantum computing, combining long-lived storage qubits with high-fidelity local operations[22,38].

**Data availability**. Relevant data are available from S.R. upon request.

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

## Acknowledgements

We thank K. Sliwa, M.J. Hatridge and A. Narla for providing the Josephson Parametric Converter (JPC), K. Chou and J.Z. Blumoff for helpful discussions, and N. Ofek for providing the logic for the field programmable gate array (FPGA) used in the control of this experiment. This research was supported by the U.S. Army Research Office (W911NF-14-1-0011). Y.Y.G. was supported by an A*STAR NSS Fellowship; P.R. by the U.S. Air Force Office of Scientific Research (FA9550-15-1-0015); C.J.A. by an NSF Graduate Research Fellowship (DGE- 1122492); S.M.G. by the National Science Foundation (DMR-1609306); L.J. by the Alfred P. Sloan Foundation and the Packard Foundation. Facilities use was supported by the Yale Institute for Nanoscience and Quantum Engineering (YINQE), the Yale SEAS cleanroom, and the National Science Foundation (MRSECDMR-1119826).

## Author contributions

S.R. and Y.Y.G. carried out measurements and data analysis. Devices were fabricated by C.W. and Y.Y.G. Experimental contributions were provided by C.W., P.R., C.J.A. and L. F. Theoretical contributions were provided by L.J. and M.M. The experiment was

designed by S.R., Y.Y.G., C.W. and R.J.S. The manuscript was written by S.R., Y.Y.G. and R.J.S. with feedback from all authors. S.M.G., M.H.D. and R.J.S. supervised the project.

## Additional information

**Competing interests:** R.J.S., M.H.D. and L.F. are equity holders and consultants at Quantum Circuits, Inc. The remaining authors declare no competing financial interests.

