## [Peer Review File · Nature Communications]

Reviewers' comments:

Reviewer #1 (Remarks to the Author):

The authors demonstrate a universal entangling gate between two logical qubits encoded in microwave cavities, using a transmon qubit to provide the necessary coupling.

The same group has previously demonstrated that similar qubit encodings allow for correction of the dominant error channels and how to perform single qubit operations on such qubits. The C-NOT entangling gate presented in this paper is now the next step towards quantum computation with such qubits.

The encoding of logical qubits in microwave resonators is particularly resource efficient and is currently one of the most promising routes towards quantum computation with error-corrected logical qubits. Therefore the results presented here will certainly be of interest to the broad audience interested in quantum computation.

It should be noted, however, that the qubit encoding chosen in this work is different from the coding the same group has used to demonstrate quantum error correction. The coding of the control qubit chosen here actually does not allow for quantum error correction, and the target and control qubit coding is different, so that in a quantum algorithm using this gate, translations between different qubit codings are required, which will be an additional source of errors. The work, therefore, will not directly be as useful as it might seem at first glance, but I find it still a significant step towards quantum computation on error-protected qubits which will certainly be stimulating for further research.

The performed measurements are very clean and well presented. I therefore recommend the paper for publication in Nat. Comms. However, I would suggest the following modifications:

- While the authors discuss to some extent the limitations of the qubit coding they choose, a nonspecialist reader might miss that point, which is quite crucial. I suggest including (parts of) the very instructive discussion at the end of the supplementary information on encodings compatible with this gate into the main paper. It would make it easier for the reader to judge the impact of the paper and more honest for non specialists.

- Fig. 2b: I would find the figure easier to read if not in the resonator rotating frame, i.e. with the resonator levels resolved in energy.

- The input density matrix should be written ρ_{in} instead of ρ_{in} which can be interpreted as matrix element $\rho_{i,n}$ in some equations.

Reviewer #2 (Remarks to the Author):

In the manuscript entitled “A CNOT gate between multiphoton qubits encoded in two cavities”, S. Rosenblum, Y.Y. Gao and colleagues report on a controlled NOT gate between two multi-photon qubits encoded in the lowest cavity mode of two separated high Q cavities.

The two superconducting cavities, referred in the manuscript as control and target cavities according to their identification in the CNOT gate, interact with a fixed frequency transmon qubit defined on a sapphire substrate. The transmon qubit is named “ancilla” and is the source of a non-linearity in the system.

The CNOT gate is implemented with a three-step process: 1) entangling the controlled multi-photon qubit with the ancilla qubit; 2) a conditional phase rotation between the ancilla and the target multi-photon qubit; 3) a final disentanglement step between the controlled and ancilla qubits.

All operations involved in the full scheme are realized via: 1) a driven sideband interaction between the control cavity and ancilla qubit; 2) a dispersive interaction between the cavities and the ancilla. The first is used to entangle the control multi-photon qubit with the $|g\rangle$ and $|f\rangle$ state of the ancilla qubit, while the second is employed in conditional phase rotations between the ancilla and the multi-photon target qubit.

The work presented here is a step forward in the approach to encode and process bits of quantum information in multi-photon systems rather than in two-level devices. The manuscript is well written, with sufficient details for a researcher to reproduce the work.

I suggest publication of the work based on the novelty and results as long as the authors can address the comments below (especially the last one related to figure 4.b)

1. Do the authors can explain the highest thermal population of the ancilla qubit compared to the other quantum elements (supplementary information, table S2)?

2. Control and target qubits are placed respectively at lower and higher frequency in respect to the ancilla qubit $|g\rangle \rightarrow |e\rangle$ transition. Is it possible to invert the order, and implement a CNOT gate with control and target at higher and lower frequencies? Can the authors comment/elaborate on this?

3. The control qubit is encoded in an even-parity Fock states, while target qubit in a Schrodinger kitten state. The first allows for error detection while the second for error correction. As stated in the manuscript and supplementary information the CNOT gate here presented is compatible with generalized kitten states, allowing for error correction in both control and target qubits. Can the authors comment on the experimental choice to have error correction encoding only on the target qubit rather than both?

4. All state fidelities reported in the manuscript are obtained by enforcing physicality, with the exception of data in figure 4.b. The caption of figure 4 motivates this choice to “avoid to the masking of certain errors”, with a reference to the Supplementary Information. Here the reason of low fidelities is described as due to errors in the Wigner tomography leading to population outside the code space. Proved this, the authors feel safely to truncate the extra levels and calculate the fidelities within the code space.

State tomography reconstruction is intrinsically affect by SPAM errors and removing “unwanted” level motivated by the low fidelity of the vacuum state does not solve it. Can the authors comment on this?

Assuming two possibilities to calculate state fidelity (imposing physicality or not), I found confusing the choice to use both in the manuscript. The authors should motivate the preference to use one over the other and use it all over the manuscript. Fidelities over 95% are reported in the main text of the manuscript but figure 4.b (from which a decay per fidelity is derived) has a maximal fidelity of 80%. This confuses the reader.

5. The authors correctly observe that “an accurate determination of the gate fidelity [of the CNOT gate] would require randomized benchmarking”. Why this measurement was not performed but it is used a rather approximate estimation of the average gate fidelity? Moreover, a two-qubit randomized benchmarking is not affected by SPAM errors (see previous observation).

6. One of the main results of the paper (the gate fidelity) is derived from the data in figure 4.b. First of all I recommend to reduce the size of the experimental dots in the figure, being extremely difficult to properly distinguish all of them and visually evaluate the quality of the fits.

I have several observations on this plot:

a) The experimental data for each initial state do not seem to align to a line. It is clearly visible an “arch” distribution, especially pronounced for the black and red triangles data. The linear fits strength this observation being the dashed-black line outside all four triangle-black experimental points with their error bar. The red-dashed fit is out of three of four experimental points. Do the authors know the reason of this data distribution and discrepancy? It would be useful to have an explanation of this effect and a fit with a function capturing it. This is important even in the view of the conclusion stated in the manuscript (next point)

b) The decay fidelity per CNOT is derived from the linear fit of the experimental data. The manuscript states that for the state $|1_L\rangle_C |X^{(\text{minus})}_L\rangle_T$ the decay in fidelity is 0.4%. The experimental points for this state are reported in figure as black-triangle, where the largest discrepancy from a linear relation is observed (previous point on this list). Between 1 and 11 CNOT operations I estimate (from the figure) a difference on state fidelity of approximately 0.1, bringing the decay in fidelity to 1%. This is inconsistent with the ancilla decay from $|f\rangle$ to $|e\rangle$ (explanation used in the manuscript to justify the 0.4%). To properly extract the decay in fidelity the authors should derive a better model to fit the experimental data.

c) Can the authors comment on the choice to apply/measure only four points (1, 4, 11, and 16 CNOT operations) per initial state? Why not measure at increasing (1 or 2) numbers of CNOT? This would have provided a better fit or insight of the data distribution.

d) The figure has a grey area defined in the caption as “simulated average slope of state fidelity imposed by ancilla decoherence”. Can the authors elaborate on this? As “average slope” I expect to see a line, while the plot shows an entire region included between two lines at different slopes.

Referee #1

While the authors discuss to some extent the limitations of the qubit coding they choose, a nonspecialist reader might miss that point, which is quite crucial. I suggest including (parts of) the very instructive discussion at the end of the supplementary information on encodings compatible with this gate into the main paper. It would make it easier for the reader to judge the impact of the paper and more honest for non specialists.

Reply: We agree with the Referee that this point deserves more emphasis in the main text. See introductory remarks and corresponding changes to the manuscript.

- Fig. 2b: I would find the figure easier to read if not in the resonator rotating frame, i.e. with the resonator levels resolved in energy.

Reply: Figure 2b has been changed to include resolved resonator levels. We indeed feel that this benefits the overall clarity of the figure.

- The input density matrix should be written ρ_{in} instead of ρ_{in} which can be interpreted as matrix element $\rho_{i,n}$ in some equations.

Reply: We modified the subscripts in order to avoid this source of confusion.

Referee #2

1) Do the authors can explain the highest thermal population of the ancilla qubit compared to the other quantum elements (supplementary information, table S2)?

Reply: The thermal population of superconducting qubits is a widespread phenomenon in the field, and is being actively investigated. Indeed, the physical temperature of the device (~20 mK) predicts a thermal population that is orders of magnitude lower than the observed one. Potential ways to reduce the population include better shielding against infrared radiation, or improved thermalization of the device. However, the detrimental effects of a high qubit temperature can be mitigated by discarding the runs in which the qubit started off in the excited state, or by initializing the qubit to the ground state at the beginning of each experimental run.

2) Control and target qubits are placed respectively at lower and higher frequency in respect to the ancilla qubit $|g\rangle\text{-}|e\rangle$ transition. Is it possible to invert the order, and implement a CNOT gate with control and target at higher and lower frequencies? Can the authors comment/elaborate on this?

Reply: It is possible to reverse the roles of control and target cavities. In practice, this only requires a change in pump frequency in order to enable the sideband interaction of the ancilla with the other cavity. However, the location of the cavity with respect to the ancilla frequency affects the ratio χ_f/χ_e . Indeed, for fixed dispersive interaction with the ancilla $|e\rangle$ -state, the interaction with the transmon in $|f\rangle$ is smaller when the cavity frequency is higher than that of the ancilla. This makes it advantageous for the target cavity frequency to be the higher-frequency one, so that the pumped control-ancilla sideband interaction is minimally affected by the target cavity photon number.

3) The control qubit is encoded in an even-parity Fock states, while target qubit in a Schrodinger kitten state. The first allows for error detection while the second for error correction. As stated in the manuscript and supplementary information the CNOT gate here presented is compatible with generalized kitten states, allowing for error correction in both control and target qubits. Can the authors comment on the experimental choice to have error correction encoding only on the target qubit rather than both?

Reply: We agree with the referee that this point is not appropriately addressed in the manuscript. See introductory remarks and corresponding changes to the manuscript.

4. All state fidelities reported in the manuscript are obtained by enforcing physicality, with the exception of data in figure 4.b. The caption of figure 4 motivates this choice to “avoid to the masking of certain errors”, with a reference to the Supplementary Information. Here the reason of low fidelities is described as due to errors in the Wigner tomography leading to population outside the code space. Proved this, the authors feel safely to truncate the extra levels and calculate the fidelities within the code space.

State tomography reconstruction is intrinsically affect by SPAM errors and removing “unwanted” level motivated by the low fidelity of the vacuum state does not solve it. Can the authors comment on this?

Assuming two possibilities to calculate state fidelity (imposing physicality or not), I found confusing the choice to use both in the manuscript. The authors should motivate the preference to use one over the other and use it all over the manuscript. Fidelities over 95% are reported in the main text of the manuscript but figure 4.b (from which a decay per fidelity is derived) has a maximal fidelity of 80%. This confuses the reader.

Reply: We regret that inconsistent definitions for state fidelities were used in the previous version of the manuscript.

The two-qubit tomography is performed by mapping the joint two-cavity parity onto the ancilla, followed by ancilla readout. This joint parity measurement has an error rate of 21%, due to imperfect readout fidelity, ancilla decoherence during the parity mapping, imperfect ancilla initialization, etc. As a result, a naïve reconstruction of the density matrix (even for vacuum in both cavities) will result in a nonphysical density matrix with nonunit trace. In addition, since a two-cavity system has an infinite-dimensional Hilbert space, we must make assumptions as to which Fock states to include in our description, and which ones to neglect. By confirming that higher photon numbers have negligible occupation probabilities consistent with noise, we can safely ignore those levels.

We agree with the Referee that SPAM errors are inherent to state tomography. Our previous decision to use normalized density matrices for calculating state fidelities, is to not divert the focus of the manuscript from the gate performance to that of state tomography. Setting the trace to unity is also problematic, since tomography errors may be correlated with gate errors. In fact, if after the gate operation the ancilla ends up in an excited state, tomography will fail, effectively resulting in a reduced trace of the reconstructed density matrix.

We acknowledge the remark of the Referee that consistent definitions should be used throughout the manuscript. For the revised manuscript we now perform the following procedure: We independently measure the parity of the vacuum state, and use this value (.79+/-0.02) to calibrate the Wigner distributions. We then perform a maximum likelihood estimation of the density matrix assuming at most two photons in the control cavity, and at most four photons in the target cavity. While we constrain the density matrix to be positive, we do not constrain its trace.

In particular, using this procedure, the state fidelity of the Bell state is now $F_{\text{out}} = 90\pm 2\%$, whereas that of the separable input state is $F_{\text{in}} = 92\pm 2\%$. The process fidelity is $89\pm 2\%$, as compared to $92\pm 2\%$ of the identity operation.

We revised the relevant section in the Supplementary Material:

For reconstructing the density matrix ρ_{CT} of the joint two-cavity system, we first measure its joint Wigner distribution. This is done by measuring the joint parity of the cavities after displacing them in their four-

dimensional phase space: (Eq. S4). Joint parity measurements are performed by a Ramsey interferometry measurement on the ancilla, which is subsequently read out. To compensate for imperfections in this procedure we calibrate the parity measurements using the value obtained for the vacuum state (0.79 +/- 0.02).

If we assume cutoffs N_C and N_T of the photon numbers in the control and target cavities, we can write ρ_{CT} as a $(N_C N_T) \times (N_C N_T)$ matrix. By measuring the joint Wigner distribution at this number of displacements or more, we can perform a maximum likelihood estimation to infer the most probable positive semi-definite Hermitian matrix ρ_{CT} . In practice, we use 6^4 different displacements, and reconstruct the two-cavity density matrix assuming fewer than six photons per cavity. We then confirm that up to the measurement accuracy there are at most four photons in the target cavity, and at most two photons in the control cavity for all measured states. This allows us then to reconstruct ρ_{CT} for this restricted 15-dimensional Hilbert space, using a now overcomplete set of data.

Since the trace of the density matrix is not constrained to unity, this method does not make the assumption that the gate operation is uncorrelated with tomography errors. Instead, failures of tomography as a result of the gate operation will show up as a reduced trace, and hence a reduced state fidelity of the final density matrix.”

5. The authors correctly observe that “an accurate determination of the gate fidelity [of the CNOT gate] would require randomized benchmarking”. Why this measurement was not performed but it is used a rather approximate estimation of the average gate fidelity? Moreover, a two-qubit randomized benchmarking is not affected by SPAM errors (see previous observation).

Reply:

This work introduces a gate between multiphoton qubits, and shows that it can achieve performance on par with traditional gates. The gate was characterized using quantum process tomography and by studying the performance under repeated gate application. We feel that an additional, more precise characterization of the gate by randomized benchmarking is outside the scope of this work, and that a more detailed study of the gate process fidelity is not required to substantiate the claims made in the manuscript. We also want to stress that, similarly to randomized benchmarking, repeated gate applications on a representative set of input states gives a result that is not affected by SPAM errors either.

6. One of the main results of the paper (the gate fidelity) is derived from the data in figure 4.b. First of all I recommend to reduce the size of the experimental dots in the figure, being extremely difficult to properly distinguish all of them and visually evaluate the quality of the fits.

Reply: We thank the referee for pointing out the insufficient clarity of Fig. 4b. The size of the markers has been reduced to allow the data points to be properly distinguished.

a) The experimental data for each initial state do not seem to align to a line. It is clearly visible an “arch” distribution, especially pronounced for the black and red triangles data. The linear fits strength this observation being the dashed-black line outside all four triangle-black experimental points with their error bar. The red-dashed fit is out of three of four experimental points. Do the authors know the reason of this data distribution and discrepancy? It would be useful to have an explanation of this effect and a fit with a function capturing it. This is important even in the view of the conclusion stated in the manuscript (next point)

Reply:

Using numerical simulations (see comment below), we expect an initially linear decrease in state fidelity of $\sim 0.5\%$ per gate application for all considered input states (except $|0\rangle_c|1\rangle_T$, which is expected to have negligible decay). For this reason, linear fits have been used. Due to the significant uncertainty of the state fidelities, our data does not claim to confirm a linear model, but statistical analysis of the results shows that it is not inconsistent with it. Rather than confirming the linearity of the decay, the data allows us to estimate decay rates given a linear model. Undoubtedly, some amount of quadratic decay behavior is to be expected due to unitary errors in the gate, such as timing errors, errors in the sideband pulse phases, etc. However, given the current accuracy of our results, we are unable to confirm whether such errors are indeed present. It is also worth pointing out that the unitary nature of these errors makes them less severe than decoherence errors, which are captured by a linear model.

b) The decay fidelity per CNOT is derived from the linear fit of the experimental data. The manuscript states that for the state $|1_L\rangle_C|X^{\text{minus}}\rangle_T$ the decay in fidelity is 0.4%. The experimental points for this state are reported in figure as black-triangle, where the largest discrepancy from a linear relation is observed (previous point on this list). Between 1 and 11 CNOT operations I estimate (from the figure) a difference on state fidelity of approximately 0.1, bringing the decay in fidelity to 1%. This is inconsistent with the ancilla decay from $|f\rangle$ to $|e\rangle$ (explanation used in the manuscript to justify the 0.4%). To properly extract the decay in fidelity the authors should derive a better model to fit the experimental data.

Reply:

When taking into consideration all of the data points, and their associated errors, we deduced a slope of $0.4\% \pm 0.3\%$ ($0.6\% \pm 0.3\%$ with the revised procedure). This value is on the order of its error, meaning that this result is consistent with, rather than a confirmation of the theoretical prediction of 0.4%. We do not feel that selectively addressing those points that indicate a higher decay rate is a consistent way of determining a decay constant.

c) Can the authors comment on the choice to apply/measure only four points (1, 4, 11, and 16 CNOT operations) per initial state? Why not measure at increasing (1 or 2) numbers of CNOT? This would have provided a better fit or insight of the data distribution.

Reply: Measuring the fidelities for repeated gate applications is a lengthy process. A single measurement takes about a millisecond. For a 36×36 density matrix, we must perform 2592 parity measurements. We repeat this process at least 500 times for sufficient averaging. Doing this for six different input states, and 1, 4, 11 or 16 gate applications, results in a total measurement time of 9 hours. Doing this for all 16 input states and for all 1 to 16 gate applications, would require an impractical total measurement time of 4 days. In addition, it is preferable not to choose a fixed interval between the points (say at multiples of four), as this could “echo away” coherent errors.

d) The figure has a grey area defined in the caption as “simulated average slope of state fidelity imposed by ancilla decoherence”. Can the authors elaborate on this? As “average slope” I expect to see a line, while the plot shows an entire region included between two lines at different slopes.

Reply: The gate operation’s susceptibility to ancilla decoherence can be simulated numerically by solving a master equation for the entire system. Without decoherence, the final state fidelities are 100%. When introducing ancilla dephasing and decay, this number is reduced by $\sim 0.5\%$ for states in which the ancilla is excited during the gate operation (all states of Fig. 4b except $|0\rangle_c|1\rangle_T$, which has no measurable

infidelity). We agree with the Referee that the gray region is inappropriate for depicting an average slope. In the revised manuscript, we replaced this with a gray line.

Reviewers' comments:

Reviewer #2 (Remarks to the Author):

I appreciate the work done by the authors in response to my criticisms. I am fully satisfied with the choice of a unique definition for the derivation of state fidelities, and with the reduced size of the experimental points in figure 4b. The new figure 4b is now easily readable. To further increase readability the authors may remove the error bars, being these on the exact size of the markers, and specify this in the caption (“error bars are the same size than the symbols”).

Caption of figure 4b should include the reason for state fidelity larger than 1.

On the other hand I am not fully convinced by the analysis of the data in figure 4b and the explanation provided.

In the response letter to the referees the authors state (referee #2, point 5):

“This work introduces a gate between multiphoton qubits, and shows that it can achieve performance on par with traditional gates.”

The comparison in performances with two-level system is reported as well at the end of the first paragraph of the manuscript:

“[...]enabling high-fidelity operations comparable to state-of-the-art gates between two-level systems.”

The reader infers that the main claims of the manuscripts are:

1. the introduction of a CNOT gate between multiphoton qubits
2. a fidelity comparable to other approaches (two-level systems)

Point (1) is very well described in the paper, and there are no doubts that this result deserves publications in this Journal.

On the other hand, if the authors claim a “high-fidelity comparable with state-of-the art gates” I will not define (response to referee #2, point 5) “[...] more precise characterization of the gate by randomized benchmarking is outside the scope of this work [...]”.

If the authors want to claim a comparable fidelity with another system it is important to use the appropriate metric (the authors state this on the manuscript: “[...] an accurate determination of the gate fidelity would require randomized benchmarking”) and provide a reference to the state-of-the art gates they compare.

If the authors prefer to relax the claims by removing the comparison sentence, the metric they decided to use is appropriate because it is fully described in the supplementary material.

Reviewer #4 (Remarks to the Author):

The paper by Rosenblum et al. describes the experimental implementation of a logical CNOT operation between two logical qubits encoded in the multiphoton states of two cavities. The authors provide reasonable evidence that this operation is implemented with high fidelity, and describe the system with an appropriate level of detail.

The authors have addressed most of the concerns of Referees #1 and #2 rather well, and there are no major issues remaining that should preclude the acceptance of the paper into Nature Communications.

That being said, I have to agree with Referee #2 that if the authors are to claim that such high-fidelities are comparable to state-of-the-art implementations in other systems, they must use higher accuracy methods to estimate these high fidelities.

The most accessible method to obtain such higher accuracy estimates would be Randomized Benchmarking (RB), as the authors acknowledge, or gate-set tomography (GST). The overhead of performing these experiments over process tomography is not large, especially in the case of RB (as RB does not attempt to reconstruct the process, just estimate the fidelity metric). Taking the experimental

numbers quoted by the authors in previous replies to referees, if each sequence of operations can be measured in approximately 1 ms, a reasonable RB experiment can measure 20 different sequence lengths, with 100 different random sequences at each length, and 50 shots per random sequence -- a total of 100k shots. To isolate the contribution from a single CNOT, this must be done twice (according to the interleaved RB protocol), leading to 200k shots. Even with overhead for recalibration and the duration of the sequence themselves, this is likely to be less than the time required to perform the 4 state tomographies reported in the paper.

While it is true that the decay of state fidelity after the repeated application of CNOTs is not affected by state preparation and measurement errors (a property RB also benefits from), it is unclear how this state fidelity decay rate relates to known and well understood error metrics. The RB decay rate, on the other hand, is well studied and known to have a simple relationship to the average fidelity of the gates under study. As Referee #2 mentions, this state fidelity decay metric the authors propose is non-standard and rather ad hoc, and is not sufficient to support the strong claim about the demonstration matching state-of-the-art implementations.

My assessment is that the authors can either (a) relax the claim that the results are comparable to state-of-the-art, or (b) provide stronger evidence in the form of 2 qubit RB or some comparable method (e.g., GST, using the open source software called pyGSTi, provided by Sandia National Labs).

Reply: Both Reviewer #4 and Reviewer #2 acknowledge the novelty and quality of our results, and Reviewer #4 states that there are "... no major issues remaining that should preclude the acceptance of the paper into Nature Communications.". Reviewer #4 also states that reasonable evidence is provided that the multiphoton CNOT operation is implemented with high fidelity. However, both Reviewers also felt that the evidence provided in the manuscript is not sufficient to justify the claim that our fidelity is comparable to state-of-the-art gates. We acknowledge that such a claim, while reasonably supported by the evidence, requires more stringent methods and more standard metrics. We have modified the claim comparing our results to state-of-the-art gates to simply a claim of high fidelity. The abstract now contains the following revised sentence: This is two orders of magnitude shorter than the decoherence time of any part of the system, enabling the gate operation to reach a high fidelity.

Reviewer #2

I am fully satisfied with the choice of a unique definition for the derivation of state fidelities, and with the reduced size of the experimental points in figure 4b. The new figure 4b is now easily readable. To further increase readability the authors may remove the error bars, being these on the exact size of the markers, and specify this in the caption ("error bars are the same size than the symbols").

Reply: We removed the error bars, and adapted the caption of fig. 4b accordingly. We feel that this change indeed benefited the clarity of the figure.

Caption of figure 4b should include the reason for state fidelity larger than 1.

Reply: The caption of fig. 4b now includes the following sentences:

The state fidelities are calibrated by the value measured for the vacuum state (see Supplementary Information). Error bars are equal in size to the symbols.

We address the remainder of Reviewer #2's remarks in the introduction of this Reply.

Reviewer #4

We acknowledge Reviewer #4's remarks, and address his concern in the revised manuscript. We provide a detailed response to the Reviewer's remarks in the introduction of this Reply.